# TRANSFER BETWEEN MODALITIES WITH METAQUERIES

## ABSTRACT

Unified multimodal models aim to integrate understanding (text output) and generation (pixel output), but aligning these different modalities within a single architecture often demands complex training recipes and careful data balancing. We introduce `MetaQueries`, a set of learnable queries that act as an efficient interface between autoregressive multimodal LLMs (MLLMs) and diffusion models. `MetaQueries` connects the MLLM's latents to the diffusion decoder, enabling knowledge-augmented image generation by leveraging the MLLM's deep understanding and reasoning capabilities. Our method simplifies training, requiring only paired image-caption data and standard diffusion objectives. Notably, this transfer is effective even when the MLLM backbone remains frozen, thereby preserving its state-of-the-art multimodal understanding capabilities while achieving strong generative performance. Additionally, our method is flexible and can be easily instruction-tuned for advanced applications such as image editing and subject-driven generation.

## 1 INTRODUCTION

The quest for unified multimodal models capable of both deep understanding (typically resulting in textual outputs) and rich generation (resulting in pixel outputs) holds immense promise. Such systems could unlock synergistic capabilities (OpenAI, 2025; Google, 2025), where understanding informs generation and vice versa. However, effectively connecting these different output modalities poses considerable challenges—*e.g.* how do we effectively transfer the latent world knowledge from the autoregressive multimodal LLM to the image generator? Although significant progress has been made, most published approaches (Ge et al., 2024; Sun et al., 2024b; Tong et al., 2024; Jin et al., 2024; Liu et al., 2024a; Team, 2024a; Xie et al., 2024; Wang et al., 2024; Wu et al., 2025; Chen et al., 2025; Dong et al., 2024; Zhou et al., 2025; Shi et al., 2024) rely on carefully tuning base multimodal LLMs (MLLMs) to handle both understanding and generation tasks. This involves complex architectural design, data/loss balancing, multiple training stages, and other complex training recipes—without these, optimizing one capability could compromise the other.

In this paper, we aim to deliver the promise of unified models via a simpler philosophy: *Render unto diffusion what is generative, and unto LLMs what is understanding.* In other words, instead of building a monolithic system from scratch, we focus on effectively transferring capabilities between state-of-the-art, pre-trained models specialized for different output modalities. To operationalize this, we keep MLLMs frozen so they can focus on what they do best—understanding—while entrusting image generation to diffusion models. We then show that even under this frozen condition, the MLLM's inherent world knowledge, strong reasoning, and in-context learning capabilities can indeed be transferred to image generation, provided the right architectural bridge is in place.

However, leveraging an MLLM—especially a frozen one—for both multimodal understanding and generation is far from straightforward. Although (frozen) LLMs have shown good performance as conditional text encoders in text-to-image generation (Zhuo et al., 2024; Xie et al., 2025; Ma et al., 2024), they are not compatible with many desired tasks in unified modeling, such as reasoning, in-context learning or producing multimodal, interleaved output. The architectural bridge we design in this work is `MetaQuery` (Figure 1). `MetaQuery` feeds a set of learnable queries directly into a frozen MLLM to extract multimodal conditions for multimodal generation. Our experiments reveal that, even without fine-tuning or enabling bi-directional attention, the frozen LLM serves as a

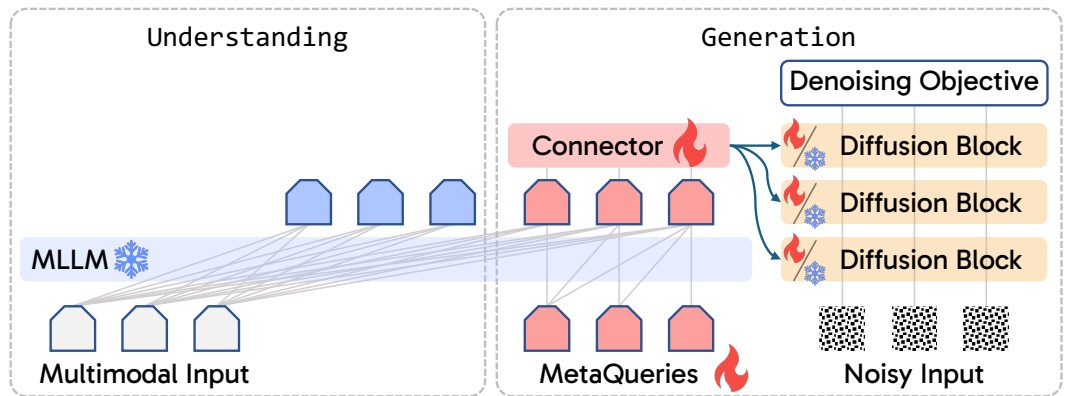

Figure 1: Overview of our model. Blue tokens maintain SOTA multimodal understanding; MetaQueries are learnable queries that directly applied to frozen MLLMs to query out conditions for generation. The model is tuned using only denoising objective with paired data. The generative diffusion models can be either frozen or further instruction-tuned for advanced generation tasks.

powerful feature resampler (Alayrac et al., 2022), producing high-quality conditions for multimodal generation. Training unified models with MetaQueries requires only a modest amount of paired image-caption data to connect these prompted conditions to any conditional diffusion model. Because the entire MLLM stays intact for understanding, the training objective remains the original denoising objective—just as efficient and stable as fine-tuning a diffusion model.

More specifically, previous unified models aim to train a single autoregressive transformer backbone to jointly model $p(\text{text},\text{pixels})$. In contrast, we choose to use a token $\rightarrow$ [transformer] $\rightarrow$ [diffusion] $\rightarrow$ pixels paradigm, which might share a high-level philosophy with the concurrent GPT-4o image generation system, as hinted at by OpenAI (2025). This approach composes the MLLM's autoregressive prior with a powerful diffusion decoder, directly leveraging the frozen MLLM's strong capability in modeling compressed semantic representations, thus avoiding the more challenging task of directly generating pixels.

To validate our approach, we conduct a series of controlled experiments, showing that MetaQuery[1] outperforms the use of a frozen MLLM purely as a conditional text encoder for image generation. Moreover, MetaQuery can match the performance of fully tuning the MLLM backbone, yet it is significantly more efficient. We also systematically investigate the training strategy, including the number of tokens and architectural configurations. With just 25M publicly available image-caption pairs, we are able to train a family of unified models that not only preserves state-of-the-art (SOTA) performance in image understanding, but also achieves SOTA-level results in text-to-image generation across multiple benchmarks.

The promise of unified modeling goes beyond handling multimodal understanding and text-to-image generation in parallel. A deeper synergy is expected—one that taps into advanced MLLM abilities like reasoning, internal knowledge, multimodal perception, and in-context learning to enhance generation. Our results show that our method draws on the frozen MLLM's commonsense knowledge, achieving SOTA visual-commonsense generation on the CommonsenseT2I benchmark (Fu et al., 2024). Our approach also harnesses the built-in reasoning and in-context learning capabilities of frozen MLLMs, producing images from complex prompts—such as generating the United States flag in response to "*The national flag of the country where Yellowstone National Park is located.*" (see Figure 7 for examples.) We also benchmark this type of world knowledge reasoning capability on WISE (Niu et al., 2025) and demonstrate SOTA performance.

Finally, by connecting, preserving, and enhancing multimodal input with MetaQueries and a frozen MLLM backbone, our model can be further instruction-tuned for advanced generation tasks such as image editing and subject-driven generation. We show that this can be achieved both efficiently and effectively using a scalable data curation pipeline that directly leverages naturally occurring image pairs from web corpora, instead of depending on human-created pairs or synthetically generated

---

[1]For simplicity, we also use MetaQuery to represent our method.

data (Brooks et al., 2023; Hu et al., 2024a; Xiao et al., 2025). This natural supervision surprisingly unlocks several new capabilities beyond subject-driven generation, such as visual association and logo design (see Figure 6 for examples).

In summary, we explore a simple yet underexplored alternative to unified multimodal modeling. Our method, `MetaQuery`, bridges frozen MLLM backbones and diffusion models. Experiments show that this framework delivers all the capabilities once thought to require MLLM fine-tuning while being much easier to train. The main results and findings in this paper include:

- With `MetaQuery` and frozen MLLM backbones, we maintain SOTA multimodal understanding performance while enabling SOTA-level multimodal generation.
- `MetaQuery` can better transfer frozen MLLMs' capabilities for reasoning- and knowledge-augmented image generation, largely outperforming existing methods.
- `MetaQuery` can extract detailed visual conditions beyond semantic similarity from frozen MLLMs, enabling image reconstruction and editing tasks.
- Our method can be easily instruction-tuned even with a frozen MLLM backbone, enabling advanced multimodal generation tasks like subject-driven generation.

## 2 RELATED WORK

**Unified understanding and generation models.** Next-token prediction has proven effective for both language (Devlin, 2019; Brown et al., 2020) and multimodal understanding (Liu et al., 2024b). Recently, the community has witnessed numerous efforts to extend the success of multimodal understanding to multimodal generation by training LLM backbones to generate images at the same time. However, unlike adapting text-only LLMs for multimodal understanding with the same next-token prediction objective (Liu et al., 2024b), multimodal generation requires distinct training objectives. SEED-X (Ge et al., 2024), Emu (Sun et al., 2024b), and MetaMorph (Tong et al., 2024) learn to regress image feature; LaVIT (Jin et al., 2024), LWM (Liu et al., 2024a), Chameleon (Team, 2024a), Show-o (Xie et al., 2024), EMU3 (Wang et al., 2024), and Janus (Wu et al., 2025; Chen et al., 2025) auto-regressively predict next visual tokens; and DreamLLM (Dong et al., 2024), Transfusion (Zhou et al., 2025) employ diffusion objectives. However, these approaches necessitate tuning LLMs for generating both modalities, naturally posing challenges in multi-task balancing.

**Unified multimodal models with learnable queries.** Learnable queries are a classic design in unified multimodal models such as DreamLLM (Dong et al., 2024) and SEED-X (Ge et al., 2024). They are effective at extracting conditions from *trainable* LLMs and work well with image feature regression or denoising objectives. However, `MetaQuery` shows that *frozen* MLLMs already excel at modeling compressed semantic representations that can be decoded into images. With lightweight tuning of the query tokens, their capacity can be fully unleashed and transferred to image generation.

**Unified multimodal models with frozen LLMs.** Several studies have explored the use of frozen LLMs for multimodal understanding and generation. LMFusion (Shi et al., 2024) trains image generation expert modules in parallel with a frozen LLM backbone to deeply fuse input conditions and denoise visual outputs. However, this approach offers limited flexibility as it shares the same architecture as specific LLM backbones and requires training a separate set of generative modules for every single LLM backbone. This not only imposes more computational burden but also restricts the ability to leverage powerful pre-trained generative models. An earlier work, GILL (Koh et al., 2023), investigates feeding learnable tokens into frozen MLLMs. It employs a regression loss for image generation, rather than directly employing the denoising objective for more efficient training. Its application is restricted to contextual image generation, and it does not systematically ablate the effects of frozen MLLMs and learnable queries.

## 3 METAQUERY

In this work, we propose `MetaQuery`, which losslessly augments understanding-only MLLMs with multimodal generation capabilities while keeping their original architecture and parameters intact. We carefully analyze the impact of applying `MetaQuery` on image generation performance. The results show that a frozen MLLM can provide strong conditions for multimodal generation.

Table 1: Study on different conditions for image generation. We observe comparable performance between learnable queries and last-layer embedding.

| Methods | Tokens | MJHQ FID ↓ | GenEval ↑ | DPG ↑ |
|---|---|---|---|---|
| LLM last-layer embedding of input tokens | - | 7.49 | 0.55 | 78.41 |
| Random queries | 64 | 8.59 | 0.35 | 54.81 |
| Learnable queries | 64 | 7.43 | 0.56 | 75.35 |
| Learnable queries | 512 | 7.34 | 0.56 | 78.43 |

Table 2: Study on strategies for adapting MLLMs. Freezing the MLLM avoids multimodal understanding degradation. This strategy achieves performance comparable to tuning the MLLM.

| Methods | Train DiT | MJHQ FID ↓ | GenEval ↑ | DPG ↑ |
|---|---|---|---|---|
| Tune MLLM | ✗ | 7.75 | 0.58 | 78.97 |
| Tune MLLM | ✓ | 6.28 | 0.61 | 79.39 |
| Freeze MLLM | ✗ | 7.43 | 0.56 | 75.35 |
| Freeze MLLM | ✓ | 6.06 | 0.61 | 76.66 |

## 3.1 ARCHITECTURE

`MetaQuery` bridges frozen MLLMs with diffusion models. We use randomly initialized learnable queries $Q \in \mathbb{R}^{N \times D}$ to query out the conditions $C$ for generation. $N$ is the number of queries and $D$ is the dimension of the queries, which is the same as the MLLM hidden dimension. For simplicity and compatibility, we continue to use causal masking for the entire sequence rather than specifically enabling full attention for $Q$. The conditions $C$ are then fed into a trainable connector to align with the input space of text-to-image diffusion models. These models can be arbitrary as long as they have a conditional input interface; we simply replace its original condition with our $C$. The whole model is trained with the original generation objective on paired data.

## 3.2 DESIGN CHOICES

The proposed architecture involves two design choices: using **learnable queries** and keeping the **MLLM backbone frozen**. We explain the reasons why we adopted these choices and how they impact performance. For all experiments, unless otherwise specified, we use the same frozen LLaVA-OneVision-0.5B (Li et al., 2024a) MLLM backbone, frozen Sana-0.6B (Xie et al., 2025) diffusion model in 512 resolution, learnable queries with $N = 64$ tokens, and a connector with a 24-layer transformer encoder. All models are trained on 25M publicly available image caption pairs for 4 epochs. We report FID score (Heusel et al., 2017) on MJHQ-30K (Li et al., 2024b) for visual quality, and GenEval (Ghosh et al., 2023) and DPG-Bench (Hu et al., 2024b) (both without prompt rewriting) for prompt alignment, respectively.

**Learnable queries.** Many models like Lumina-Next (Zhuo et al., 2024), Sana (Xie et al., 2025), and Kosmos-G (Pan et al., 2024) use the (M)LLM's last-layer embedding of input tokens as image generation conditions. However, this approach is not ideal for unified models as it is incompatible with many desired tasks in unified modeling, such as reasoning, in-context learning or producing multimodal, interleaved output (We provide additional results on highly related reasoning-augmented image generation task in Section 5.4, where learnable queries largely outperform LLM's last-layer embedding). As shown in Table 1, using learnable queries with just $N = 64$ tokens achieves image generation quality comparable to that of utilizing the last-layer embedding of input tokens. Additionally, since the last-layer embedding setting naturally comes with a longer sequence length, we also tested learnable queries with $N = 512$ tokens, which further improves performance and even outperforms the last-layer embedding approach.

**Frozen MLLM.** Existing unified models train MLLMs to jointly model $p(\texttt{text}, \texttt{pixels})$, resulting in a more complicated training process and even downgraded understanding performance. `MetaQuery` keeps the original MLLM architecture and parameters intact to preserve SOTA understanding capabilities. However, for multimodal generation, a key concern is whether `MetaQuery`'s performance with significantly fewer tunable parameters would be substantially worse than full MLLM tuning. In Table 2, frozen MLLMs achieve comparable performance to full MLLM tuning, with slightly lower prompt alignment but slightly improved visual quality. Training DiT can further improve performance for both settings. This suggests that `MetaQuery` is another possible training strategy, one that is simpler yet effective, as an alternative to fine-tuning the entire MLLM.

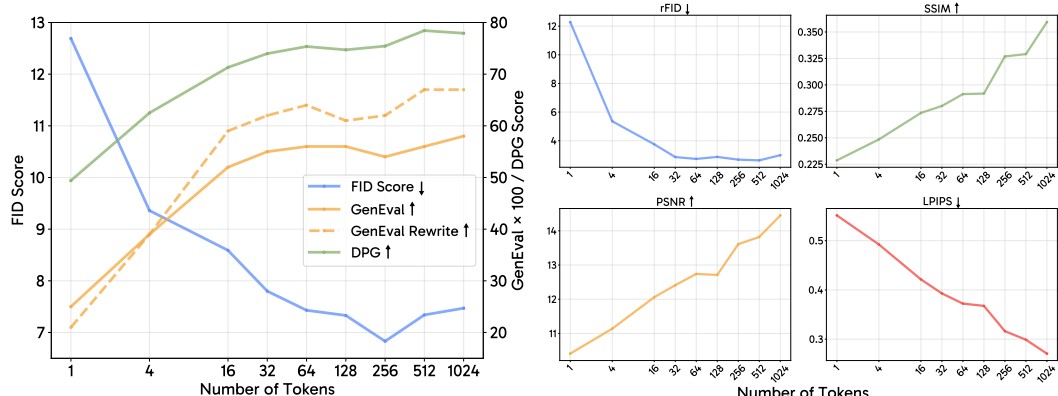

Figure 2: Study on the scaling of token numbers. As the number of tokens increases, text-to-image prompt alignment (left) and image reconstruction results (right) consistently improve.

## 3.3 TRAINING RECIPE

Based on insights from our design choices, we further examine key training options for the two main components of `MetaQuery`: the number of tokens and connector design. Unless otherwise specified, all experiments in this section use the same setup as described in Section 3.2.

**Number of tokens.** Many works (Wu et al., 2023; Pan et al., 2024; Ge et al., 2024) have employed learnable queries for condition extraction. However, they either set the number of tokens to the image decoder's input length (e.g., $N = 77$ for Stable Diffusion v1.5 (Rombach et al., 2021)), or use an arbitrary number like $N = 64$ without further study. Given that modern diffusion models like Sana (Xie et al., 2025) support variable-length conditions, determining the optimal number of tokens is crucial. In Figure 2, we provide a careful study of the number of tokens and observe promising scalability of `MetaQueries`. For text-to-image generation, visual quality begins to converge after 64 tokens, while more tokens consistently improves prompt alignment. This effect is stronger for long captions, where GenEval with rewritten prompts grows more rapidly. For image reconstruction, more tokens consistently enhance reconstruction quality (visual samples in Appendix A.1). We set $N = 256$ in later experiments as a good balance between performance and efficiency.

**Connector design.** The connector is another key component in `MetaQuery`. We use the same architecture as Qwen2.5 (Team, 2024b) LLM but enable bi-directional attention. We explore two designs: Projection–Encoder (Proj-Enc) and Encoder–Projection (Enc-Proj). Proj-Enc first projects conditions to the diffusion decoder dimension, and then aligns them to the diffusion decoder space, while Enc-Proj aligns them in the MLLM hidden dimension before projection. As the results show in Appendix A.2, Enc-Proj achieves better performance with fewer parameters.

## 4 MODEL TRAINING

We train `MetaQuery` in two stages: pre-training and instruction tuning, Both keeping MLLMs frozen while fine-tuning other modules. We adopt three different MLLM backbones: Base (LLaVA-OneVision 0.5B (Li et al., 2024a)), Large (Qwen2.5-VL 3B (Bai et al., 2025)), and X-Large (Qwen2.5-VL 7B (Bai et al., 2025)). All models use $N = 256$ tokens and a 24-layer Enc-Proj connector. For image generation heads, we test two different models: Stable Diffusion v1.5 (Rombach et al., 2021) and Sana-1.6B (Xie et al., 2025).

**Pre-training.** We pre-train our model on 25M publicly available image-caption pairs for 8 epochs with a learning rate of 1e-4 and a global batch size of 4096. The learning rate follows a cosine decay schedule with a 4,000-step warmup period before gradually decreasing to 1e-5.

**Instruction tuning.** Furthermore, in this work, we rethink the data curation process for instruction tuning in image generation. All current methods rely on expert models to generate target images from source images and instructions (Ge et al., 2024; Xiao et al., 2025; Hu et al., 2024a). However, this approach is limited in scalability and may introduce biases, as the available expert models cover only a narrow range of image transformations.

Inspired by MagicLens (Zhang et al., 2024), we construct instruction-tuning data from naturally occurring image pairs in web corpora. These corpora contain rich multimodal contexts with interleaved text and images on related subjects or topics. The pairs capture meaningful and diverse relationships, from visual similarity to subtle semantic connections (Figure 3). Such data provides strong supervision for instruction tuning. Based on this observation, we design a pipeline that mines these pairs and uses MLLMs to generate open-ended instructions describing their relationships. First, we collect grouped images from

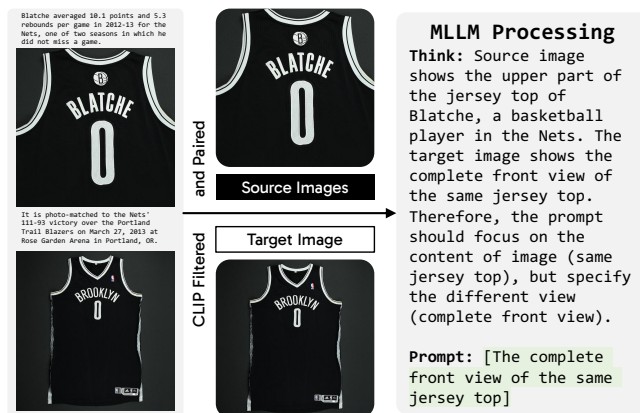

Figure 3: Instruction tuning data curation pipeline. Images are first grouped from web corpora based on caption similarity, then paired to construct instruction data using an MLLM.

mmc4 (Zhu et al., 2023) core fewer-faces subset, where each image is accompanied by a caption. We then cluster them using SigLIP (Zhai et al., 2023) (up to 6 images per group, with a similarity threshold of 0.5). In each group, the image with lowest average similarity to the others is designated as the target, while the remaining images serve as source images. This process yields 2.4M image pairs, describing how to transform the source images into the target image (see Appendix B for the prompt). We experimented with instruction-tuning our Base model on the proposed 2.4M dataset for 3 epochs, using the same learning rate schedule as in pre-training and a batch size of 2048.

## 5  EXPERIMENTS

In this section, we first evaluate `MetaQuery` on various multimodal understanding and text-to-image generation benchmarks (Section 5.1). We demonstrate that `MetaQuery` can be trained to reconstruct input images, and this reconstruction capability can be easily transferred to perform image editing (Section 5.2). Furthermore, we show that `MetaQuery` can be instruction-tuned to perform zero-shot subject-driven generation (Section 5.3). By leveraging our approach for collecting instruction tuning data from naturally existing image pairs, we also reveal that `MetaQuery` can unlock novel capabilities like visual association and logo design (also in Section 5.3). Additionally, we demonstrate that `MetaQuery` can transfer the internal knowledge and reasoning capabilities of the frozen MLLM, overcoming common failures exhibited by other generation models (Section 5.4).

### 5.1  IMAGE UNDERSTANDING AND GENERATION

As shown in Table 3, our model family demonstrates strong capabilities across both understanding and generation tasks. Benefiting from the flexible training approach that allows us to leverage arbitrary SOTA frozen MLLMs, models of all sizes exhibit competitive performance on understanding benchmarks (Fu et al., 2023; Liu et al., 2023; Li et al., 2023a; Yue et al., 2024; Yu et al., 2023).

For image generation, `MetaQuery` achieves SOTA visual quality on MJHQ-30K (Li et al., 2024b). Since `MetaQuery` works with frozen MLLMs, it naturally connects to an arbitrary number of diffusion models. As the base Sana-1.6B (Xie et al., 2025) model is already fine-tuned on aesthetic data, we adopt frozen Stable Diffusion v1.5 (Rombach et al., 2021) for COCO evaluation. After adapting to powerful MLLMs, we obtain a COCO FID of 8.69, surpassing base model performance. This also establishes a new SOTA among SD v1.5-based unified models, including MetaMorph (Tong et al., 2024) (11.80) and Emu (Sun et al., 2024b) (11.66) (more comparison in Appendix C). Qualitative results of text-to-image generation can be found in Figure 4. In terms of prompt alignment, `MetaQuery` also achieves competitive performance on GenEval (Ghosh et al., 2023) and DPG-Bench (Hu et al., 2024b), beating all diffusion-based methods including Transfusion (Zhou et al., 2025) and JanusFlow (Ma et al., 2025). However, there remains a gap with Janus-Pro (Chen et al., 2025), which auto-regressively predicts image tokens. We suggest that this gap may be due to the

Table 3: Results on multimodal understanding and generation. We report COCO FID with frozen Stable Diffusion v1.5 (Rombach et al., 2021), and other metrics with fine-tuned Sana 1.6B (Xie et al., 2025). † denotes rewritten prompts; ‡ results are tested by us under the same settings.

| Methods | Base (M)LLM | MME-P | MMB | SEED | MMMU | MM-Vet | COCO FID ↓ | MJHQ FID ↓ | GenEval ↑ | DPG ↑ | WISE ↑ |
|---|---|---|---|---|---|---|---|---|---|---|---|
| Emu | LLaMA 13B | - | - | - | - | - | 11.66 | - | - | - | - |
| DreamLLM | Vicuna 7B | - | - | - | - | 36.6 | 8.46 | - | - | - | - |
| SEED-X | LLaMA-2 13B | 1457.0 | 70.1 | 66.5 | 35.6 | 43.0 | 17.88 | 19.53 | 0.51 | 75.51 | - |
| Chameleon | From Scratch 7B | - | - | - | 22.4 | 8.3 | 26.74 | - | 0.39 | - | - |
| Show-o-512 | Phi-1.5 1.3B | 1097.2 | - | - | 26.7 | - | 9.24 | 15.18 | 0.68 | - | 0.35 |
| VILA-U | LLaMA-2 7B | 1401.8 | - | 59.0 | - | 33.5 | - | 7.69 | - | - | 0.31 |
| Emu3 | From Scratch 7B | - | 58.5 | 68.2 | 31.6 | 37.2 | 12.80 | - | 0.66† | 80.60 | 0.39 |
| MetaMorph | LLaMA-3 8B | - | 75.2 | 71.8 | - | - | 11.80 | - | - | - | - |
| TokenFlow-XL | Qwen-2.5 14B | 1551.1 | 76.8 | 72.6 | 43.2 | 48.2 | - | - | 0.63† | 73.38 | - |
| Transfusion | From Scratch 7B | - | - | - | - | - | 8.70 | - | 0.63 | - | - |
| LMFusion | LLaVA-Next 8B | 1603.7 | 72.1 | 72.5 | 41.7 | - | 8.20 | - | - | - | - |
| Janus | DeepSeek-LLM 1.5B | 1338.0 | 69.4 | 63.7 | 30.5 | 34.3 | 8.53 | 10.10 | 0.61 | - | 0.23 |
| JanusFlow | DeepSeek-LLM 1.5B | 1333.1 | 74.9 | 70.5 | 29.3 | 30.9 | - | 9.51 | 0.63 | 80.09 | 0.18 |
| Janus-Pro-1B | DeepSeek-LLM 1.5B | 1444.0 | 75.5 | 68.3 | 36.3 | 39.8 | - | 14.33‡ | 0.73 | 82.63 | 0.26 |
| Janus-Pro-7B | DeepSeek-LLM 7B | 1567.1 | 79.2 | 72.1 | 41.0 | 50.0 | - | 13.48‡ | 0.80 | 84.19 | 0.35 |
| MetaQuery-B | LLaVA-ov 0.5B | 1238.0 | 58.5 | 66.6 | 31.4 | 29.1 | 8.91 | 6.28 | 0.74† | 80.04 | 0.46 |
| MetaQuery-L | Qwen2.5-VL 3B | 1574.3 | 78.6 | 73.8 | 53.1 | 63.2 | 8.87 | 6.35 | 0.78† | 81.10 | 0.55 |
| MetaQuery-XL | Qwen2.5-VL 7B | 1685.2 | 83.5 | 76.9 | 58.6 | 66.6 | 8.69 | 6.02 | 0.80† | 82.05 | 0.55 |

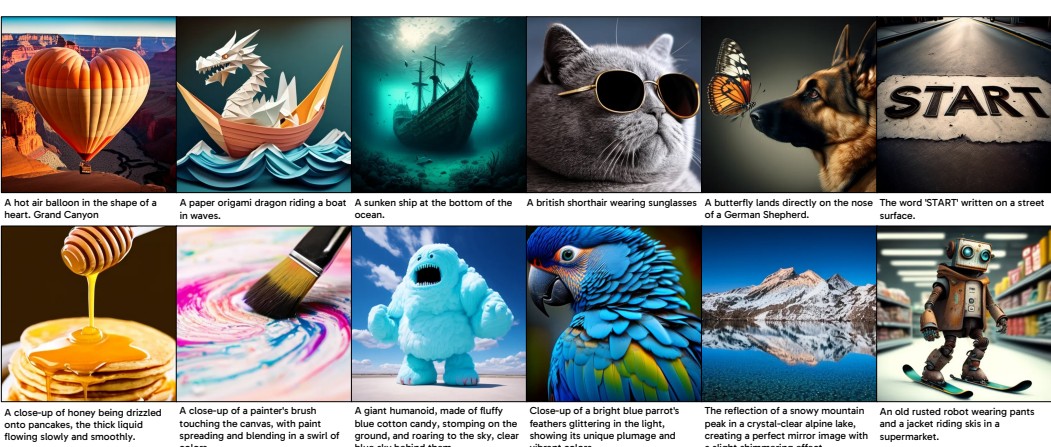

Figure 4: Qualitative results of text-to-image generation. Prompts are from PartiPrompt (Yu et al., 2022), Sana (Xie et al., 2025) and Movie Gen Bench (Polyak et al., 2024).

different failure modes of diffusion models and auto-regressive models: diffusion models usually fail to correctly follow the prompt, while auto-regressive models may suffer from more visual artifacts. We test the MJHQ-30K FID score of Janus-Pro and find that, in terms of visual quality and artifact control, MetaQuery is significantly better than Janus-Pro (see Appendix E for visual comparison).

We find that MetaQuery is effective at transferring the internal knowledge and reasoning capabilities of the frozen MLLM, achieving significantly better performance on WISE (Niu et al., 2025). We elaborate on this in detail in Section 5.4. Finally, we find that scaling up the size of frozen LLMs further improves generation quality and prompt alignment. We also discuss the impact of employing different LLM variants of the same size in Appendix D. MetaQuery provides a simple and principled way to leverage the most advanced multimodal LLMs within a unified modeling framework.

## 5.2 IMAGE RECONSTRUCTION AND EDITING

We demonstrate that MetaQuery can be easily fine-tuned for image reconstruction tasks with a frozen MLLM (see Appendix F for visual samples and details). Our fine-tuned MetaQuery-B can achieve competitive performance, matching the best existing open-source model Emu2 (Sun et al., 2024a), and even comparable to GPT-4o (OpenAI, 2025). In Figure 5, we show that MetaQuery can effectively transfer its image reconstruction capability to perform image editing. We keep the MLLM frozen and fine-tune our pre-trained Base model for only 1,000 steps on publicly available image editing data, the fine-tuned model is already capable of general image editing scenarios.

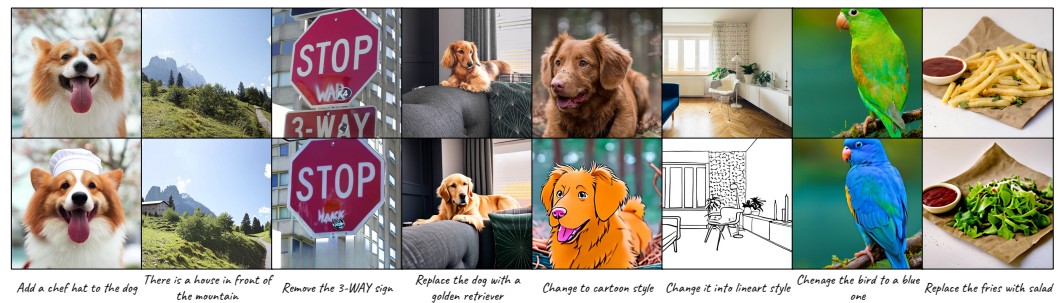

Figure 5: Image editing results. It is transferable from reconstruction with lightweight fine-tuning.

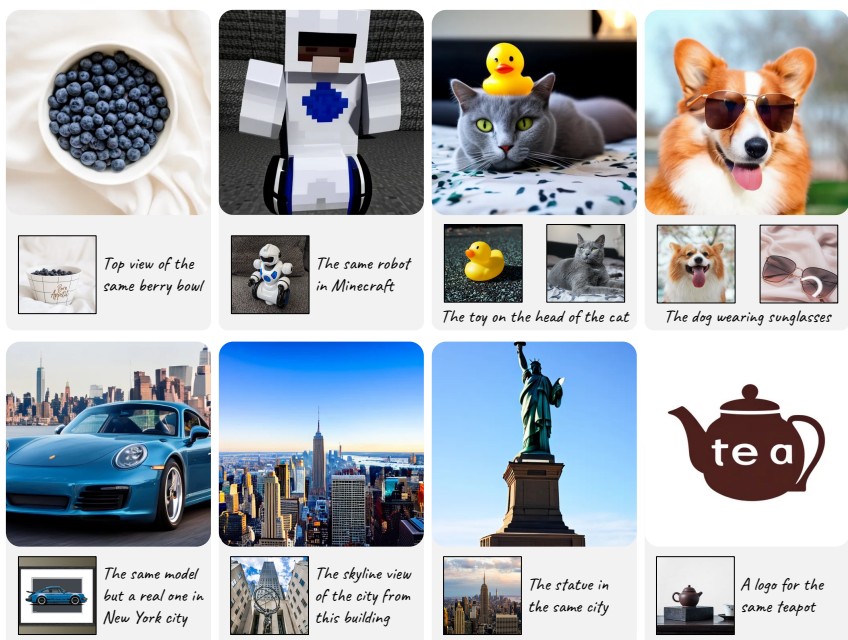

Figure 6: Qualitative results for instruction tuning. Instruction-tuned `MetaQuery` shows strong subject-driven capability (top) and can reason over multimodal input to generate images (bottom).

## 5.3 INSTRUCTION TUNING

We show that after being instruction-tuned on the proposed 2.4M dataset in Section 4, `MetaQuery` can achieve impressive zero-shot subject-driven generation performance, producing coherent results even with multiple highly customized subjects (the first row of Figure 6). We further quantitatively evaluate subject-driven generation on DreamBench (Ruiz et al., 2023), where our `MetaQuery-B-Instruct` achieves SOTA performance (see Appendix G for detailed results). Using various supervision signals, the instruction-tuned `MetaQuery-B` model surprisingly unlocks novel capabilities like visual association and logo design that go beyond copy-pasting (the second row of Figure 6). For example, in the second case, the model recognizes the input image of Rockefeller Center and imagines the view of New York City from the top of the Rockefeller Center.

## 5.4 REASONING- AND KNOWLEDGE-AUGMENTED GENERATION

We show that the learnable queries can effectively transfer capabilities of the frozen LLM. This enables the model to better understand and follow complex prompts, including those requiring real-world knowledge and reasoning. As shown in Figure 7, for the left knowledge-augmented generation cases, `MetaQuery-L` can leverage world knowledge from the frozen MLLM and reason through the input question to generate the correct answer. For the right commonsense knowledge cases from CommonsenseT2I (Fu et al., 2024), the LLM provides better commonsense knowledge and enables `MetaQuery` to generate images that are consistent with the facts.

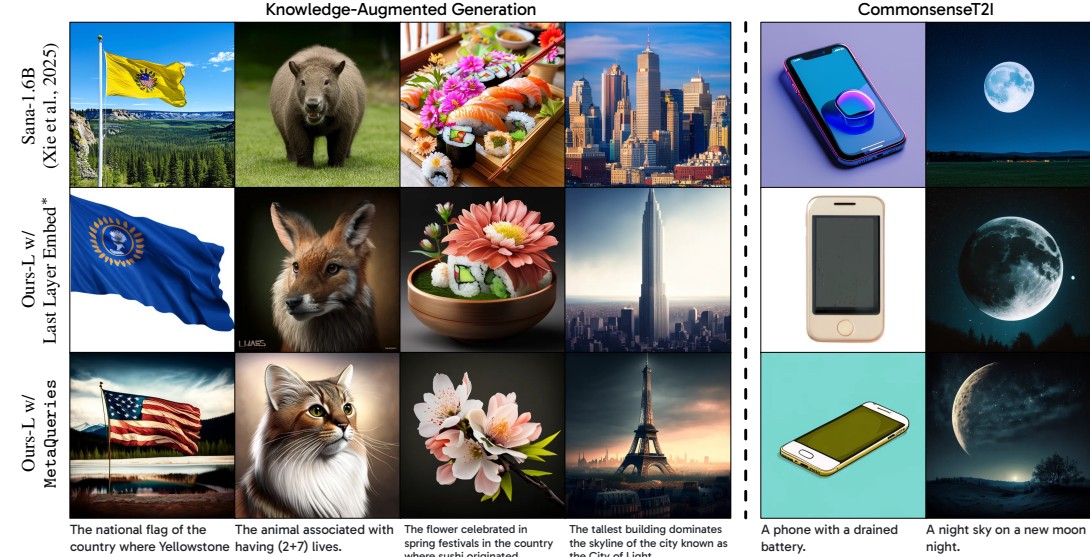

Figure 7: `MetaQuery` effectively leverages frozen MLLMs for reasoning- and knowledge-augmented generation. * indicates using LLM last-layer embeddings of input tokens as condition, which cannot activate in-context learning and still fails in most cases.

Table 4: Quantitative comparison on reasoning- and knowledge-augmented image generation. `MetaQuery` largely improves over the base model and the LLM last-layer embedding baseline.

| Methods | WISE ↑ | CommonsenseT2I w/ Neg. Prompt ↑ |
|---|---|---|
| Sana 1.6B | 0.50 | 43.33 |
| LLM last-layer embedding of input tokens | 0.48 | 52.83 |
| `MetaQueries`-L | 0.55 | 57.67 |

To evaluate `MetaQuery`'s world knowledge reasoning capability, we employ the WISE (Niu et al., 2025) benchmark, which resembles the knowledge-augmented generation cases in Figure 7. As shown in Table 3, `MetaQuery` achieves SOTA performance, significantly outperforming all other models. Notably, prior approaches failed to leverage MLLMs effectively, lagging behind text-to-image models like Sana. `MetaQuery` stands as the first unified model to successfully transfer these advanced capabilities of MLLMs to image generation and to surpass SOTA text-to-image models. We also demonstrate `MetaQuery`'s SOTA commonsense reasoning capability on the CommonsenseT2I (Fu et al., 2024) (see Appendix H for detailed results).

As discussed before in Table 1, one key advantage of `MetaQuery` over LLM's last-layer embeddings is that `MetaQuery` is integrated natively with the LLM, which better supports reasoning and in-context learning. The last-layer approach treats the decoder-only LLM as a text encoder, limiting its in-context learning capabilities. As shown in Figure 7, it still fails in cases where the LLM must first reason over input questions before generating corresponding images, which requires in-context learning beyond what text encoders typically provide. This performance gap is also quantitatively confirmed in Table 4, where `MetaQuery` significantly outperforms the last-layer embedding approach on both WISE and CommonsenseT2I benchmarks.

## 6 CONCLUSION

We presented `MetaQueries`, a simple interface connecting MLLMs (for understanding) and diffusion decoders (for generation), effective even when the MLLM is frozen. This approach yields state-of-the-art understanding and generation performance with straightforward implementation. By enabling transfer between modalities, `MetaQueries` successfully channels MLLM knowledge and reasoning into multimodal generation. While effective, we hypothesize that bridging the remaining gap to leading proprietary systems may primarily involve further data scaling. We hope `MetaQueries` provides a powerful, accessible baseline for future unified multimodal model development.

## REPRODUCIBILITY STATEMENT

To ensure reproducibility, we provide our data curation code, training code, and detailed config files in the supplementary material. A complete description of the data curation steps can be found in Section 4 and Appendix B.

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

In this paper, we employ Large Language Models (LLMs) to perform grammar checking as part of the writing process. They were not involved in research ideation.

## A DETAILS ON TRAINING RECIPE STUDY

### A.1 NUMBER OF TOKENS

We present the visual samples of different numbers of tokens in Figure 8. When using $N = 1$ token, the reconstructed images are only semantically aligned with the input. As the number of tokens increases, subtle details become progressively more consistent with the input.

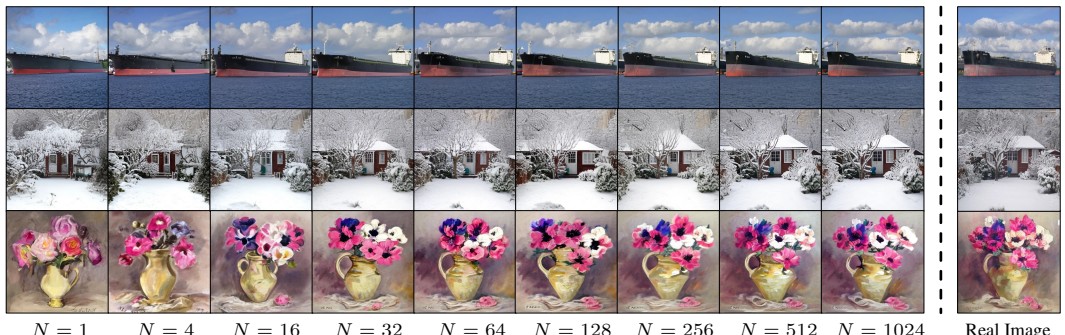

$N = 1$     $N = 4$     $N = 16$     $N = 32$     $N = 64$     $N = 128$     $N = 256$     $N = 512$     $N = 1024$     Real Image

Figure 8: Visaul samples for image reconstruction with different numbers of tokens.

### A.2 CONNECTOR DEISGN

We provide the detailed results of the connector design study in Table 5. The results show that Enc-Proj is both more powerful and more efficient.

Table 5: Study on connector design. Aligning the conditions first in the same dimension as the MLLM hidden states (Enc-Proj) is more effective and parameter-efficient.

| Architecture | Layers | Dims | Params | Rel. Wall Time | MJHQ FID $\downarrow$ | GenEval $\uparrow$ | DPG $\uparrow$ |
|---|---|---|---|---|---|---|---|
| Proj-Enc | 6 | 2304 | 517M | 1.06x | 7.80 | 0.53 | 73.37 |
| Proj-Enc | 24 | 2304 | 2046M | 1.23x | 7.41 | 0.51 | 73.75 |
| Enc-Proj | 6 | 896 | 84M | 1x | 7.73 | 0.49 | 71.39 |
| Enc-Proj | 24 | 896 | 316M | 1.06x | 7.43 | 0.56 | 75.35 |

## B DATA CURATION DETAILS

For the data curation part, we use `Qwen/Qwen2-VL-7B-Instruct`[2] as our MLLM, The system prompt we are using is:

---

[2] `https://huggingface.co/Qwen/Qwen2-VL-7B-Instruct`

Based on the provided of one or multiple source images, one target image, and their captions, create an interesting text prompt that can be used with the source images to generate the target image.

This prompt should include:

- one general and unspecific similarity shared with the source images (same jersey top, similar axe, similar building, etc).
- all differences that only the target image has.

This prompt should NOT include:

- any specific details that would allow generating the target image independently without referencing the source images.

Remember the prompt should be concise and short. The generation has to be done by combining the source images and text prompts.

## C    COMPARISON WITH BASE MODELS

It is non-trivial to fully preserve, or even enhance, the base models' SOTA performance in both understanding and generation. To address this, we propose keeping the MLLM backbone frozen while only tuning the proposed `MetaQuery`, thereby maintaining the base models' understanding capabilities. As shown in Table 6, `MetaQuery` also achieves improvements in image generation, as evidenced by its gains over the base SD v1.5 (Rombach et al., 2021) model. We choose the frozen SD v1.5 to provide a fair comparison with other unified models.

Table 6: Comparison with other SD v1.5 Rombach et al. (2021)-based unified models on text-to-image performance. We use frozen SD v1.5 for a fair comparison. `MetaQuery` is the only one that does not cause performance degradation of the base model.

| Methods | COCO FID ↓ | GenEval ↑ | DPG ↑ |
|---|---|---|---|
| Stable Diffusion v1.5 (SD v1.5) | 9.20 | 0.43 | 63.18 |
| Emu (Sun et al., 2024b) | 11.66 | - | - |
| MetaMorph (Tong et al., 2024) | 11.80 | - | - |
| MetaQuery-XL w/ Frozen SD v1.5 | 8.69 | 0.45 | 66.64 |

## D    COMPARISON OVER DIFFERENT LLM BACKBONES

We further discuss the impact of employing different LLM backbones for `MetaQuery` in Table 7. Concretely, we carefully select a family of backbone models: pre-trained LLM (Qwen2.5-3B), instruction-tuned LLM (Qwen2.5-3B-Instruct), and instruction-tuned MLLM (Qwen2.5-VL-3B-Instruct). Both instruction-tuned models are initialized with the first pre-trained model checkpoint. Experimental results show that instruction tuning can achieve better (multimodal) understanding capabilities. However, the improvements are orthogonal to image generation performance when employed to provide multimodal generation conditions.

Table 7: Comparison across different LLM backbones. Image generation capability is mostly orthogonal to multimodal understanding capability.

| LLM Backbones | MJHQ FID ↓ | GenEval ↑ | DPG ↑ | CommonsenseT2I ↑ |
|---|---|---|---|---|
| Qwen2.5-3B | 6.20 | 0.79 | 81.34 | 56.00 |
| Qwen2.5-3B-Instruct | 6.36 | 0.79 | 81.12 | 54.33 |
| Qwen2.5-VL-3B-Instruct | 6.35 | 0.78 | 81.10 | 57.67 |

# E  QUALITATIVE COMPARISON OF TEXT-TO-IMAGE GENERATION

We provide a qualitative comparison with Janus-Pro-7B (Chen et al., 2025) on MJHQ-30K (Li et al., 2024b) in Figure 9. We can see that `MetaQuery-XL` follows the prompt better and generates more visually appealing images than Janus-Pro-7B.

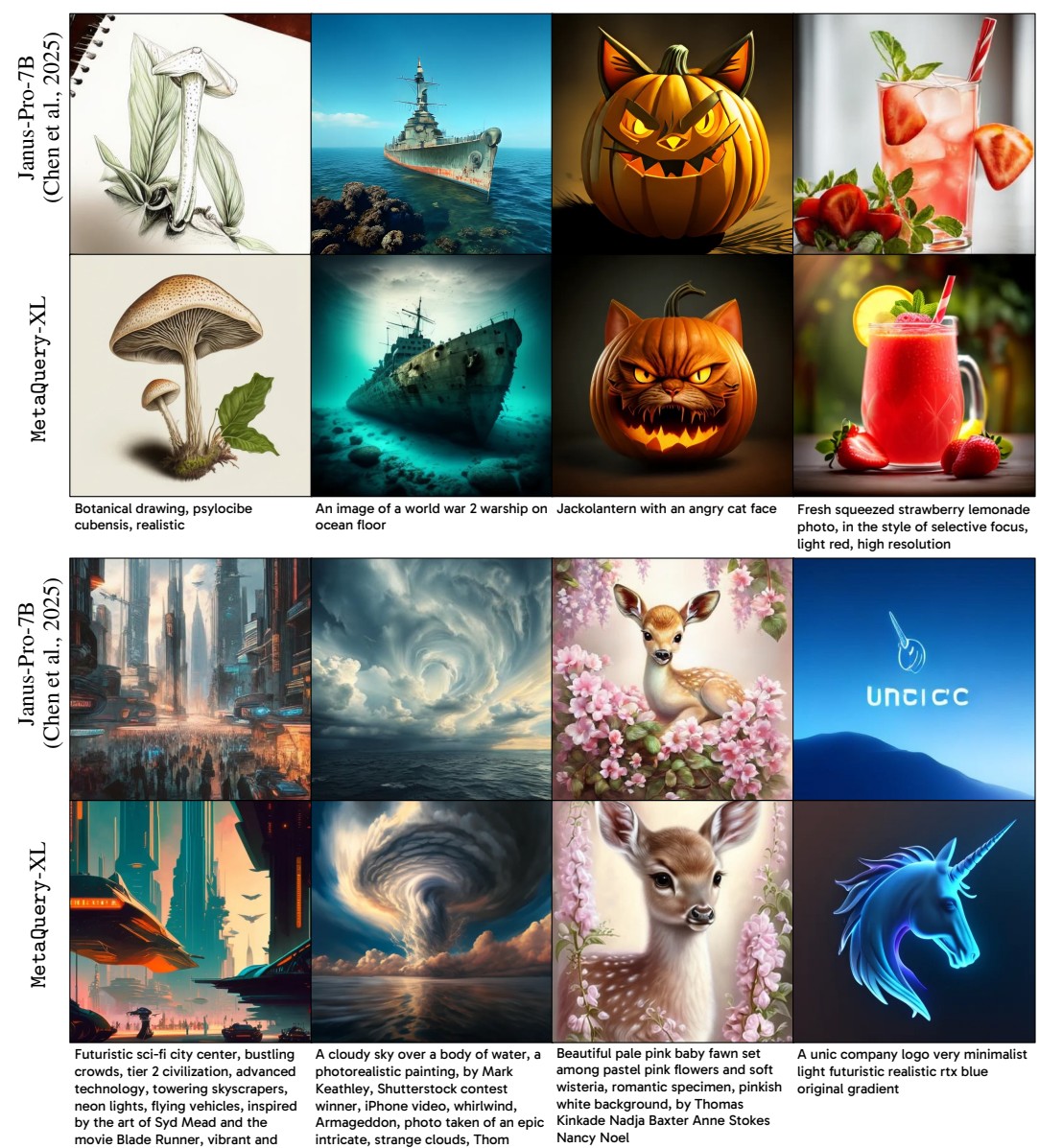

Figure 9: Qualitative comparison with Janus-Pro-7B (Chen et al., 2025) on MJHQ-30K.

# F  IMAGE RECONSTRUCTION

`MetaQuery` is able to reconstruct images from frozen MLLMs. Since we are using an MLLM for multimodal perception, besides the standard text-to-image objective, we can also employ an image reconstruction objective at the same time. In Table 8, we show that a mix of both objectives can enable image reconstruction capabilities without being generally harmful to the T2I performance.

Table 8: Study on training objectives. Image reconstruction objective can be mixed with text-to-image objective to enable image reconstruction capabilities without harming visual quality and prompt alignment.

| Objective | Rel. Wall Time | MJHQ FID ↓ | GenEval ↑ | DPG ↑ |
|---|---|---|---|---|
| Text-to-Image | 1.0x | 7.43 | 0.56 | 75.35 |
| Image Reconstruction | 2.79x | 27.42 | 0.32 | 68.36 |
| Mix | 2.61x | 8.27 | 0.54 | 76.53 |

We show the visual samples of image reconstruction in Figure 10, we compare our fine-tuned `MetaQuery-B` with existing diffusion autoencoders from various unified models, which reconstruct images from predicted visual features. Since these unified models are not explicitly fine-tuned for image reconstruction, their results are directly decoded from the vision encoder's output. Remarkably, even under this more constrained setup, our fine-tuned `MetaQuery-B` can still achieve competitive performance, matching the best existing open-source model Emu2 (Sun et al., 2024a). When compared with GPT-4o (OpenAI, 2025), our model also achieves comparable quality.

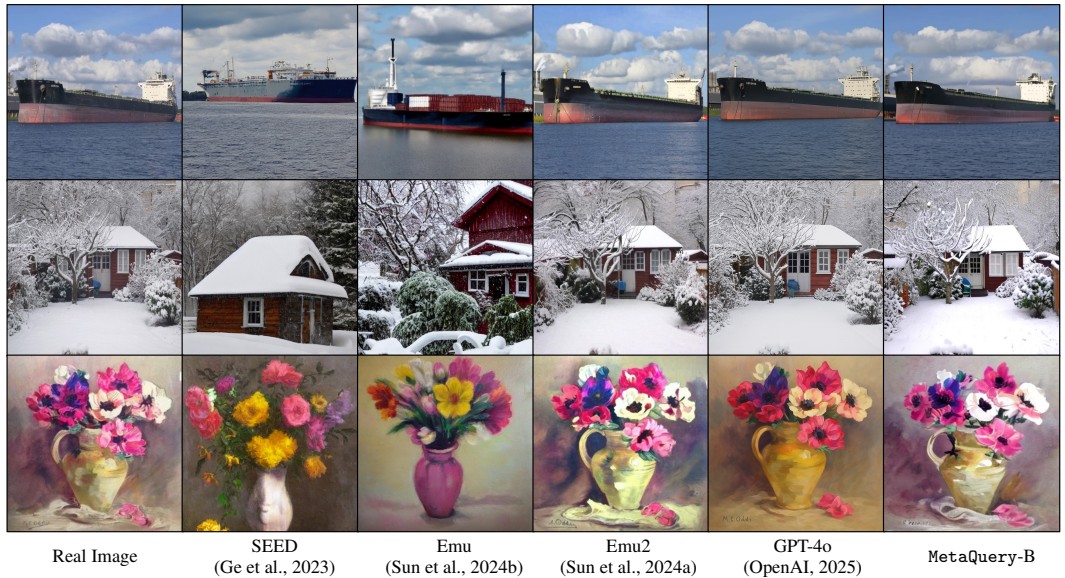

| Real Image | SEED (Ge et al., 2023) | Emu (Sun et al., 2024b) | Emu2 (Sun et al., 2024a) | GPT-4o (OpenAI, 2025) | MetaQuery-B |

Figure 10: Image reconstruction results. Results of SEED, Emu, and Emu2 are from Sun et al. (2024a).

# G  QUANTITATIVE RESULTS OF SUBJECT-DRIVEN GENERATION

To quantitatively evaluate `MetaQuery`-Instruct's subject-driven image generation capability, we follow DreamBooth (Ruiz et al., 2023) by adopting DINO, CLIP-I, and CLIP-T scores to evaluate our model on the DreamBench (Ruiz et al., 2023) dataset. As shown in Table 9, our `MetaQuery`-B-Instruct model achieves SOTA performance, outperforming existing models like Kosmos-G (Pan et al., 2024) that are explicitly trained on constructed substitution tasks for subject-driven generation.

Table 9: Subject-driven generation results on DreamBench (Ruiz et al., 2023).

| Methods | DINO Score↑ | CLIP-I Score↑ | CLIP-T Score↑ |
|---|---|---|---|
| Real Images (Oracle) | 0.774 | 0.885 | - |
| *fine-tuning* | | | |
| Textual Inversion (Gal et al., 2023) | 0.569 | 0.780 | 0.255 |
| DreamBooth (Ruiz et al., 2023) | 0.668 | 0.803 | 0.305 |
| BLIP-Diffusion (Li et al., 2023b) | 0.670 | 0.805 | 0.302 |
| *zero-shot & test time tuning free* | | | |
| Re-Imagen (Chen et al., 2023) | 0.600 | 0.740 | 0.270 |
| BLIP-Diffusion (Li et al., 2023b) | 0.594 | 0.779 | 0.300 |
| Kosmos-G (Pan et al., 2024) | 0.694 | 0.847 | 0.287 |
| MetaQuery-B-Instruct | 0.737 | 0.852 | 0.301 |

# H ADDITIONAL RESULTS OF REASONING- AND KNOWLEDGE-AUGMENTED GENERATION

We further quantitatively evaluate MetaQuery's commonsense reasoning capability on the CommonsenseT2I benchmark (Fu et al., 2024) in Table 10. For simplicity, we use CLIP (Radford et al., 2021) as the evaluator following their original implementation. Results show that MetaQuery significantly improves the performance of the base Sana model, achieving SOTA performance.

Table 10: Comparison of visual commonsense reasoning on CommonsenseT2I (Fu et al., 2024).

| Methods | w/o Neg. Prompt | w/ Neg. Prompt |
|---|---|---|
| DALL-E 3 (Ramesh et al., 2021) w/ rewrite | 40.17 | N/A |
| SD-3-medium (Esser et al., 2024) | 26.17 | 47.17 |
| FLUX.1-dev (Labs, 2024) | 24.50 | 22.50 |
| Sana-1.6B (Xie et al., 2025) | 25.17 | 43.33 |
| MetaQuery-B | 27.33 | 51.50 |
| MetaQuery-L | 28.83 | 57.67 |

