# OpenReview forum: "Transfer between Modalities with MetaQueries"
_ICLR.cc/2026/Conference — Submitted to ICLR 2026_

### Official Review · Reviewer_hJkF · 2025-10-28

**Soundness:** 3
**Presentation:** 3
**Contribution:** 2
**Rating:** 2
**Confidence:** 4

**Summary:**

The paper presents an interesting approach that freezes the LLM backbone solely for understanding tasks, while delegating generation tasks to image-generative models. This design simplifies data balancing and removes the need for complex training procedures.

**Strengths:**

1.The paper introduces MetaQuery, a framework that leverages the capabilities of frozen MLLMs.

2.The proposed method achieves state-of-the-art performance in both multimodal understanding and image generation tasks.

3.The approach also shows potential for extension to other image-related applications, such as image editing, through appropriate fine-tuning.

**Weaknesses:**

1. The concept does not appear entirely novel, as similar ideas have been explored in prior works such as Seed-X, MetaMorph, and Next-GPT. Could the authors clarify the main differences between their approach and these methods?

2. It would strengthen the paper if the authors could include additional experiments on subtasks that more directly assess the benefits of a unified model, such as interleaved image-text generation or other multimodal interaction tasks.

**Questions:**

See the weakness.

---

> ### Author Response · Authors · 2025-11-22
> **Official Rebuttal (1/3)**
>
> Dear Reviwer hJkF,
>
> Thanks a lot for your feedback. We would like to address your concerns one by one.
>
> **Q1: Lack of novelty**
>
> **A1:** We agree with the reviewer that (i) connecting an LLM with a diffusion model and (ii) using learnable queries are not novel by themselves. Our goal is not to propose a brand-new concept in that sense, but to demonstrate that **a different, significantly more efficient paradigm is feasible for building unified models**. Instead of training the LLM to produce image-aligned embeddings, we keep a MLLM fully frozen and extract information from its hidden states using learnable queries. This approach is similar in spirit to prompt tuning [1]. In our experiments, this lightweight tuning preserves the original understanding capabilities of the MLLM while still delivering everything needed for generation (text-to-image generation, image reconstruction, image editing, subject-driven generation, and knowledge-augmented generation). If this behavior were already standard practice, we would expect to see more unified systems that keep the backbone completely frozen. However, existing unified models almost always modify the LLM, in order to joint train understanding and generation within a single backbone.
>
> **Why are we using learnable queries?**
>
> To work with frozen MLLMs, we note that learnable queries have many advantages over other methods:
>
> 1. Compared to input tokens’ last hidden states, query embeddings offer several benefits:
>    1. They provide flexible control over the budget, as the number of query tokens can be adjusted to suit different tasks or resources.
>    2. They support stronger in-context learning and reasoning capabilities (see Table 7 in UniLip [2]).
>    3. They are well-suited for streaming and long-context scenarion. e.g., a chat history spanning thousands of tokens, where it becomes impractical to condition image generation on all input tokens. In contrast, query embeddings offer a compressed, fixed-length representation that is more interpretable for diffusion models.
> 2. Compared with MoT: Some concurrent works [3, 4] train image generation expert modules in parallel with a frozen LLM backbone to deeply fuse input conditions and guide the denoising process. However, this setup offers limited flexibility because it shares the same architecture of a specific LLM backboneand requires training a separate set of generative modules for every single LLM backbone. This also increases computational overhead. For instance, in LMFusion [3], the additional parameters used for generation total is nearly 7B, equivalent to the size of the MLLM itself. In contrast, MetaQuery only requires about 1.6B parameters, which is the size of the diffusion model. This design also makes it easier to integrate powerful, pre-trained generative models without modifying the backbone.
>
> **Who are generating images?**
>
> Compare to models that are training MLLMs, our approach changes the direction of information flow from a *Push* paradigm to a *Pull* paradigm.
>
> 1. The *Push* Paradigm (MetaMorph, Seed-X, Next-GPT, DreamLLM [5, 6, 7, 8]): Prior works operate on a *Push* mechanism, **LLMs learns to generate images**. They train the LLM to generate image-aligned embeddings and *push* them to the generator. The LLM is forced to alter its output space to accommodate generation, which often requires substantial parameter updates and risks catastrophic forgetting of text capabilities. An evidence of this forgetting risks is that all unified models in below Table achieves worse reasoning performance than T2I models (with frozen text encoders). This highlights our efforts to better transfer the capabilities of LLMs to image generation.
>
> 2. The *Pull* Paradigm (Ours - MetaQueries): In contrast, our method establishes a *Pull* mechanism, **diffusion models are generating images**. We treat the MLLM as a frozen, parametric knowledge base. Instead of forcing it to output visual signals, we use MetaQueries to actively probe and *pull* rich semantic features directly from its internal hidden states via self-attention. Generation remains entirely in the diffusion model, and the MLLM does not change its output distribution or parameters. By *pulling* directly from internal representations rather than *pushing* an output, we preserve the MLLM’s original capabilities intact (Frozen Backbone), and better transfer them to image generation.
>
> **Table1. Comparison on reasoning augmented generation on WISE**
>
> | **Model**             | **WISE** |
> | --------------------- | -------- |
> | **Dedicated T2I**     |          |
> | FLUX.1-dev            | 0.50     |
> | PixArt-Alpha          | 0.47     |
> | SD-v1-5               | 0.32     |
> | SD-3.5-large          | 0.46     |
> | **Unify MLLM**        |          |
> | Emu3                  | 0.39     |
> | JanusFlow-1.3B        | 0.18     |
> | Janus-Pro-7B          | 0.35     |
> | show-o-512            | 0.35     |
> | vila-u-7b-256         | 0.31     |
> | MetaQuery-XL          | **0.55** |

---

> ### Author Response · Authors · 2025-11-22
> **Official Rebuttal (2/3)**
>
> **A1-cnt**: We continue with the detailed comparison with the required methods.
>
> **Compared with SEED-X [5]**
>
> Our models share a similar design with SEED-X in that both use query embeddings for image generation. However, as discussed above, SEED-X relies on the LLM to generate visual representations using an MSE loss. As shown in Emu2 [9] and RAE [10], visual representations can be deterministically decoded into pixels. This mapping from visual representations to pixels is essentially a reconstruction task, not generation. In this setup, predicting visual representations means the MLLM itself does most of the generation process. In contrast, MetaQuery just *pull* information from the MLLMs. The frozen MLLM has nothing to do with generation; instead, generation is entirely handled by the diffusion model.
>
> **Compared with MetaMorph [6]**
>
> Similarly, MetaMorph predicts visual representations in an autoregressive manner, rather than using query embeddings. In MetaMorph, the authors focus on exploring the synergy that arises when a single backbone performs both next-text-token prediction (understanding) and visual representation prediction (generation). This paradigm is fundamentally different from MetaQuery, which is designed to separate understanding and generation.
>
> **Compared with Next-GPT [7]**
>
> Next-GPT also trains the MLLM to predict visual representations using an MSE loss. Beyond the differences mentioned above, Next-GPT applies learnable queries through small adapter modules. These query tokens only interact with the encoder’s final outputs and do not influence the internal computations of the backbone, as they merely resample a static representation. In contrast, MetaQueries are more akin to prompt tuning [1] on frozen LLMs. We append learnable queries directly to the input sequence and process them using the LLM’s native causal self-attention. As a result, MetaQueries remain part of the context window throughout the entire forward pass. They aggregate not only static features but also the dynamic reasoning trajectory of the LLM.

---

> ### Author Response · Authors · 2025-11-22
> **Official Rebuttal (3/3)**
>
> **Q2: Subtasks that more directly assess the benefits of a unified model**
>
> **A2:** We respectfully clarify that the core research question of this paper is: *Can we **transfer** the complex reasoning and world knowledge of a frozen SOTA MLLM to image generation?* To this end, we selected subtasks such as Knowledge-Augmented Generation and Reasoning-Intensive Prompts that directly assess this question.
>
> However, we wish to highlight that without introducing any architectural changes, our model can be post-trained to support multi-turn, interleaved interactions. Specifically, following OmniGen2 [11], we can introduce an `<img>` token and fine-tune only this token on an interleaved corpus. When the model predicts the `<img>` token, we append the metaqueries to generate an image, encode it back into the context, and resume the conversation.
>
> However, for these Interleaved geneation, it's hard to assess the benefits of a unified model since itself doesn't directly benefit geneartion or understanding. However, we agree this is a promising direction. Recent work such as ThinkMorph [12] has explored interleaved chain-of-thought generation, demonstrating the benefits of unified models in significantly enhancing text reasoning capabilities. Crucially, ThinkMorph is based on the Bagel [4] model, which also uses a frozen MLLM during unified model pre-training. This suggests that freezing MLLMs at the pre-training stage does not necessarily limit their capabilities in the post-training phase.
>
> That said, this line of extensive investigation is beyond the current scope of our work, as it is heavy and remains underexplored. If the reviewer is interested, we would be happy to provide a brief qualitative demo of Self-CoT similar to what was done in Bagel to illustrate multimodal interaction in a revised version.
>
> **Reference:**
>
> [1] Lester, B., Al-Rfou, R., & Constant, N. (2021). The power of scale for parameter-efficient prompt tuning. *arXiv preprint arXiv:2104.08691*.
>
> [2] Tang, H., Xie, C., Bao, X., Weng, T., Li, P., Zheng, Y., & Wang, L. (2025). Unilip: Adapting clip for unified multimodal understanding, generation and editing. *arXiv preprint arXiv:2507.23278*.
>
> [3] Shi, W., Han, X., Zhou, C., Liang, W., Lin, X. V., Zettlemoyer, L., & Yu, L. (2024). LMFusion: Adapting Pretrained Language Models for Multimodal Generation. *arXiv preprint arXiv:2412.15188*.
>
> [4] Deng, C., Zhu, D., Li, K., Gou, C., Li, F., Wang, Z., ... & Fan, H. (2025). Emerging properties in unified multimodal pretraining. *arXiv preprint arXiv:2505.14683*.
>
> [5] Ge, Y., Zhao, S., Zhu, J., Ge, Y., Yi, K., Song, L., ... & Shan, Y. (2024). Seed-x: Multimodal models with unified multi-granularity comprehension and generation. arXiv preprint arXiv:2404.14396.
>
> [6] Tong, S., Fan, D., Li, J., Xiong, Y., Chen, X., Sinha, K., ... & Liu, Z. (2025). Metamorph: Multimodal understanding and generation via instruction tuning. In Proceedings of the IEEE/CVF International Conference on Computer Vision (pp. 17001-17012).
>
> [7] Wu, S., Fei, H., Qu, L., Ji, W., & Chua, T. S. (2024, July). Next-gpt: Any-to-any multimodal llm. In Forty-first International Conference on Machine Learning.
>
> [8] Dong, R., Han, C., Peng, Y., Qi, Z., Ge, Z., Yang, J., ... & Yi, L. (2023). Dreamllm: Synergistic multimodal comprehension and creation. arXiv preprint arXiv:2309.11499.
>
> [9] Sun, Q., Cui, Y., Zhang, X., Zhang, F., Yu, Q., Wang, Y., ... & Wang, X. (2024). Generative multimodal models are in-context learners. In *Proceedings of the IEEE/CVF Conference on Computer Vision and Pattern Recognition* (pp. 14398-14409).
>
> [10] Zheng, B., Ma, N., Tong, S., & Xie, S. (2025). Diffusion Transformers with Representation Autoencoders. *arXiv preprint arXiv:2510.11690*.
>
> [11] Wu, C., Zheng, P., Yan, R., Xiao, S., Luo, X., Wang, Y., ... & Liu, Z. (2025). OmniGen2: Exploration to Advanced Multimodal Generation. *arXiv preprint arXiv:2506.18871*.
>
> [12] Gu, J., Hao, Y., Wang, H. W., Li, L., Shieh, M. Q., Choi, Y., ... & Cheng, Y. (2025). ThinkMorph: Emergent Properties in Multimodal Interleaved Chain-of-Thought Reasoning. *arXiv preprint arXiv:2510.27492*.

---

> ### Comment · Reviewer_hJkF · 2025-11-28
>
> Firstly, thank you for your detailed response. I have read it very carefully.
>
> I will increase my score accordingly when I can. However, I would like to further discuss the definition of a unified model. From my perspective, simply using a connector to combine two existing modules may not be the most effective approach to exploring how to design a better unified model. Architectures such as Bagel[1] or BLIP-3o[2] might represent more promising directions. This concern has been the primary reason for my reservations. Although your rebuttal to the first weakness partially addresses this issue, I remain somewhat uncertain about the overall motivation of the work. It would be helpful if you could provide additional clarification regarding your motivation and your perspective on the proposed architecture.
>
> I also acknowledge your claim that the method is easy to generalize and use. However, I noticed that the connector includes up to 2B parameters in the Appendix, which is somewhat beyond my expectations.
>
> For the second weakness, your response addresses my concern. Given the limited time, I do not expect you to provide new experimental results; additional clarification and explanation would already be helpful. Finally, I have read the comments and scores provided by the other reviewers, and I will determine my final score after the discussion phase.
>
> Best regards.
>
> [1] Deng, Chaorui, et al. "Emerging properties in unified multimodal pretraining, 2025." URL https://arxiv. org/abs/2505.14683.
>
> [2] Chen J, Xu Z, Pan X, et al. Blip3-o: A family of fully open unified multimodal models-architecture, training and dataset[J]. arXiv preprint arXiv:2505.09568, 2025.

---

> ### Author Response · Authors · 2025-12-04
>
> Dear Reviewer hJkF,
>
> Thank you very much for your reply! We're excited to further discuss this with you, as your question touches directly on the main motivation of our work.
>
> In our view, a unified model should be defined by its functionality, not its architecture. Specifically, we define a unified model as one that supports both multimodal understanding and multimodal generation. This can be achieved in multiple ways, and it doesn’t necessarily require training a single Transformer backbone (as in EMU, SEED-X, Next-GPT, show-o, or MetaMorph). The motivation for using a shared Transformer backbone is to encourage synergy between the two tasks. While prior work like MetaMorph demonstrates that understanding can benefit generation, we haven't seen compelling evidence in the literature that generation improves understanding. In fact, in practice, multimodal understanding performance often degrades when co-trained with image generation.
>
> As stated in our introduction, MetaQuery is designed to transfer the capabilities of LLMs to image generation (i.e., understanding helps generation), while preserving the LLM’s original understanding capabilities. This simple yet effective design highlights the advantages of unified multimodal modeling, while circumventing its common drawbacks (generation harms understanding).
>
> Similarly, concurrent works such as BLIP-3o and BAGEL also explore strategies that keep the LLM frozen. From this perspective, our motivation is very similar. Notably, BLIP-3o adopts exactly the MetaQuery architecture to extract conditioning signals for the image generator. The only difference is the design of the image generator: BLIP-3o denoises CLIP features, whereas we denoise VAE features. Additionally, while MetaQuery may appear to “connect two existing modules” due to our use of a pre-trained image generator to accelerate training, our architecture remains fully compatible with image generators trained from scratch, as in BLIP-3o.
>
> BAGEL trains image generation expert modules in parallel with a frozen LLM backbone. However, this setup offers limited flexibility, as it depends on the architecture of a specific LLM backbone and requires training a separate set of generative modules for each one. This also increases computational overhead. For instance, in BAGEL, the additional parameters used for generation total 7B, equal to the size of the LLM itself. In contrast, MetaQuery-XL requires only about 1.6B additional parameters, corresponding to the size of the diffusion model. This design also facilitates the integration of powerful, pre-trained generative models without modifying the backbone. Notably, MetaQuery-XL also better transfers knowledge and reasoning capabilities from LLMs than BAGEL (0.55 vs. 0.52 on WISE), while using significantly fewer parameters and data (25M vs. 1600M image-caption pairs). This advantage is further supported in the current paper [2] (see Table 4), which shows that query-based methods can transfer knowledge from LLMs more effectively than MoT.
>
> Thank you also for your question about the connector design. In our initial experiments, we used a large Transformer-based connector to link the T5 text encoder with the Stable Diffusion model [1]. However, we later found that, by fine-tuning the diffusion model and adding a LayerNorm before injecting the conditioning (to stabilize gradients), we could effectively use a much lighter MLP connector. In fact, our results show that this setup can even outperform the heavier Transformer-based connector. We will update the paper with these results in the camera-ready version.
>
> **Table1. Comparison with MLP connector**
>
> | Connector                            | GenEval | DPG   |
> | ------------------------------------ | ------- | ----- |
> | Qwen 2.5 VL 3B w/ 24-layer connector | 0.78    | 81.10 |
> | Qwen 2.5 VL 3B w/ MLP (10M params)   | 0.79    | 82.69 |
>
>
>
> **Reference:**
>
> [1] Qin, C., Yu, N., Xing, C., Zhang, S., Chen, Z., Ermon, S., ... & Xu, R. (2023). Gluegen: Plug and play multi-modal encoders for x-to-image generation. In *Proceedings of the IEEE/CVF international conference on computer vision* (pp. 23085-23096).
>
> [2] Niu, Y., Jin, W., Liao, J., Feng, C., Jin, P., Lin, B., ... & Yuan, L. (2025). Does Understanding Inform Generation in Unified Multimodal Models? From Analysis to Path Forward. *arXiv preprint arXiv:2511.20561*.

---

### Official Review · Reviewer_7rvH · 2025-10-28

**Soundness:** 4
**Presentation:** 4
**Contribution:** 2
**Rating:** 4
**Confidence:** 4

**Summary:**

This paper addresses the challenge of creating unified multimodal models that excel at both understanding (text output) and generation (pixel output) without performance degradation. The authors propose **MetaQueries**, a set of learnable tokens that act as an efficient interface between a **frozen** autoregressive multimodal LLM (MLLM) and a diffusion model.

**Strengths:**

1. The core idea of MetaQueries as a "plug-and-play" interface is a significant strength, promoting modularity in a field dominated by monolithic models.
2. The paper's most impressive contribution is the *proof* (via Figure 7 and Table 4) that MetaQueries are not just a simple conditioning mechanism but a functional bridge for transferring high-level MLLM capabilities (world knowledge, reasoning) to the generation process. SOTA on the WISE benchmark confirms this.
3. The "Tune vs. Freeze MLLM" and "MetaQueries vs. Last-Layer Embedding" ablations are critical, well-executed, and strongly support the paper's central claims.

**Weaknesses:**

1. The paper honestly notes (Section 5.1) that its method lags behind autoregressive (AR) visual-token models like Janus-Pro on prompt-alignment benchmarks (GenEval, DPG). While the authors provide a qualitative defense (better visual quality, Appendix E), this remains a quantitative gap. It would be beneficial to discuss if this is a fundamental limitation of the diffusion-decoder approach or if it could be closed with more data/tuning.
2. In Section 3.1, the paper states, "we continue to use causal masking for the entire sequence". This is slightly ambiguous. A clearer explanation of the exact attention mask between the `[Multimodal Input]` and `[MetaQueries]` would be helpful. For instance, do the queries attend to the input, but the input cannot attend to the queries?
3. The trainable "connector" is a key component, but it's not deeply analyzed. Appendix A.2 explores two designs, but the impact of connector depth/complexity on performance vs. efficiency is not fully explored. How much of the "alignment" is handled by the queries versus this 24-layer transformer?

**Questions:**

1. Following up on Weakness #1: Do you believe the prompt-alignment gap (vs. AR models like Janus-Pro) is a fundamental trade-off for the superior visual quality of diffusion, or could this gap be closed, perhaps by scaling the 25M image-caption pre-training data or further tuning the connector?
2. Following up on Weakness #2: Could you please clarify the exact attention mechanism? Given a sequence `[Input Tokens, MetaQuery Tokens]`, what is the attention mask? Is it a standard causal mask over the entire concatenated sequence?
3. In Table 2, the "Freeze MLLM" setting achieves a *better* (lower) FID score than "Tune MLLM" (e.g., 6.06 vs 6.28 when training DiT). This is counter-intuitive, as one might expect tuning to help. Do you have a hypothesis for why *not* tuning the MLLM leads to slightly better visual quality?
4. The instruction-tuning data pipeline (Section 4) is very clever. How sensitive is the model's performance to the MLLM-generated instruction? For example, did you find that variations in the system prompt (Appendix B) led to significant differences in the model's final editing/subject-driven capabilities?

---

> ### Author Response · Authors · 2025-11-22
> **Official Rebuttal**
>
> Dear Reviewer 7rvH,
>
> Thanks a lot for your valuable questions. We would like to address your concerns one by one.
>
> **Q1: AR v.s. Diffusion**
>
> **A1:** We believe this question touches on the broader debate between autoregressive (AR) image generation and diffusion models. This remains an open area, as there is little literature providing a controlled comparison between the two paradigms. That said, many state-of-the-art image generation models such as FLUX and Stable Diffusion 3.5 are diffusion-based, which suggests there are no fundamental limitations to using diffusion decoders. The discussion in our paper is intended to highlight empirical observations from our experiments, where we noticed different failure modes between AR and diffusion models. Specifically, AR models tend to achieve better prompt alignment, while diffusion models are generally better at producing high-fidelity images. Additionally, it's important to note that Janus-Pro was trained on approximately 150M samples, whereas our model was trained on only around 25M. We believe that with more data and compute, the performance gap can be closed.
>
> **Q2: The mask mechanism**
>
> **A2:** It is a standard causal mask over the entire sequence. We append the query tokens after all input tokens, so the queries attend to the input, but the input cannot attend to the queries.
>
> **Q3: Deeply Analyze Connector**
>
> **A3:** Thank you for your question. In our initial experiments, we followed the approach of connecting T5 text encoders with the Stable Diffusion model using a large transformer-based connector [1]. Now we discovered that if we fine-tune the diffusion model and add a LayerNorm before injecting the conditioning into the pretrained diffusion model to stablize the gradient, we can successfully use a much lighter MLP connector. In fact, we show that this setup can even outperform heavier transformer-based connectors. We will update results with this setup in revision.
>
> **Table1. Comparison with MLP connector**
>
> | Connector                            | GenEval | DPG   |
> | ------------------------------------ | ------- | ----- |
> | Qwen 2.5 VL 3B w/ 24-layer connector | 0.78    | 81.10 |
> | Qwen 2.5 VL 3B w/ MLP (10M params)   | 0.79    | 82.69 |
>
> Now, with only a lightweight MLP connector, we are better able to understand how the model works. The diffusion models are trained to align with the MLLM outputs and primarily handle alignment. In contrast, the queries are mainly responsible for information extraction and resampling from the MLLM.
>
> **Q4: Why freezing MLLM achieve better FID score**
>
> **A4:** We believe the difference is minor and may not be sufficient to show that freezing is fundamentally better. However, keeping the text encoder frozen is a standard practice in training text-to-image diffusion models. Empirically, we observe two main advantages may lead to better FID score:
>
> 1. Preservation of pretrained capability: The frozen MLLMs have been pretrained on trillions of tokens. Fine-tuning the entire backbone on our much smaller text-to-image dataset may lead to suboptimal results.
> 2. Training stability: Keeping the input conditions for the diffusion model frozen makes it easier for the model to learn to generate images reliably. This results in a more stable training dynamic when aligning two pretrained modules: the MLLM and the diffusion model.
>
> **Q5: System prompt’s impact**
>
> **A5:** The instructions generated by MLLMs need to be carefully curated, similar to how caption quality significantly affects text-to-image generation, instruction quality greatly influences the model’s ability to follow instructions. As for the system prompt used to generate these instructions, we didn’t tune it too much, as it works quite well when provided with in-context examples. In such cases, the MLLM follows the examples very well. You can refer to the in-context examples provided in the code for more details.
>
> **Reference:**
>
> [1] Qin, C., Yu, N., Xing, C., Zhang, S., Chen, Z., Ermon, S., ... & Xu, R. (2023). Gluegen: Plug and play multi-modal encoders for x-to-image generation. In *Proceedings of the IEEE/CVF international conference on computer vision* (pp. 23085-23096).

---

### Official Review · Reviewer_NM2y · 2025-10-29

**Soundness:** 3
**Presentation:** 3
**Contribution:** 2
**Rating:** 6
**Confidence:** 4

**Summary:**

This paper proposed to unify multimodal understanding and generation by combining bespoke pretrained models. Compared to existing unified LLMs, this work adopted a trainable connector to act as a bridge between pre-trained MLLMs and the diffusion decoder. Learnable queries as input to fuse vision-language information from the pre-trained MLLMs as input conditions for diffusion decoding. Extensive experimental results demonstrated the effectiveness of the proposed approach.

**Strengths:**

1. The paper is well-organized, and the figures are well-prepared.
2. This approach achieves state-of-the-art results on the existing multimodal understanding and generation benchmarks.
3. The reasoning and knowledge-augmented generation looks interesting.

**Weaknesses:**

1. My biggest concern is the size of the connector, which is extremely large when adopting Qwen2.5-VL 3B and 7B as the base MLLMs. The connector size may be larger than the diffusion decoder, which makes the claim that this transfer is effective even when the MLLM backbone remains frozen less convincing.

2. Given the above point, the paradigm of MetaMorph may be more effective. This raises another concern: what if we take the MLLM's output tokens as direct input for the connector? Would it be more effective, or would there be fewer learnable parameters required to make this transfer? Additional experimental results are required.

**Questions:**

See weaknesses. I will adjust the rating according to the authors' response.

**Details Of Ethics Concerns:**

NA.

---

> ### Author Response · Authors · 2025-11-22
> **Official Rebuttal**
>
> Dear Reviewer NM2y,
>
> Thanks a lot for your valuable questions. We will address your concerns one by one.
>
> **Q1: The connector is too large**
>
> **A1:** Thank you for your question. In our initial experiments, we followed the approach of connecting T5 text encoders with the Stable Diffusion model using a large transformer-based connector [1]. Now we discovered that if we fine-tune the diffusion model and add a LayerNorm before injecting the conditioning into the pretrained diffusion model to stablize the gradient, we can successfully use a much lighter MLP connector. In fact, we show that this setup can even outperform heavier transformer-based connectors. We will update results with this setup in revision.
>
> **Table1. Comparison with MLP connector**
>
> | Connector                            | GenEval | DPG   |
> | ------------------------------------ | ------- | ----- |
> | Qwen 2.5 VL 3B w/ 24-layer connector | 0.78    | 81.10 |
> | Qwen 2.5 VL 3B w/ MLP (10M params)   | 0.79    | 82.69 |
>
> **Q2: Directly taking MLLM’s output tokens as input for the connector**
>
> **A2:** Thank you for your suggestion. If we understand correctly, by “MLLM's output tokens,” you’re referring to autoregressively predicting hidden states from LLMs like MetaMorph. The issue arises because MetaMorph is explicitly aligned with SigLIP image features, providing a ground truth for teacher forcing during training. In contrast, MetaQuery does not directly predict SigLIP image features from MLLMs, but instead trains diffusion models. As a result, we lack a ground truth for teacher forcing during training. This necessitates heavy token-by-token prediction across all tokens during training, which is less efficient. If our understanding is incorrect, we’d be happy to clarify further or follow up with additional experimental results.
>
> **Reference:**
>
> [1] Qin, C., Yu, N., Xing, C., Zhang, S., Chen, Z., Ermon, S., ... & Xu, R. (2023). Gluegen: Plug and play multi-modal encoders for x-to-image generation. In *Proceedings of the IEEE/CVF international conference on computer vision* (pp. 23085-23096).

---

### Official Review · Reviewer_Z3r9 · 2025-11-02

**Soundness:** 2
**Presentation:** 3
**Contribution:** 2
**Rating:** 4
**Confidence:** 4

**Summary:**

The authors propose MetaQueries, a simple and light module that bridges the frozen LLM and a diffusion model, to boost the multimodal understanding and generation. MetaQueries is a series of learnable tokens that fed into the LLM to learn latent conditions to align to the conditional space of diffusion models. While the authors conduct experiments over multiple benchmarks, I think the contribution mostly comes from the engineering, rather than the technical novelty.

**Strengths:**

- text is easy to follow
- method is simple and efficient, by inserting learnable tokens, the training process only needs to fine-tune diffusion model.

**Weaknesses:**

- limited novelty: the main component, MetaQueries, is essentially a form of learnable prompts / queries, similar to prior adapters such as Q-Former
- lack of theoretical analysis: no deeper analysis on why or when frozen MLLM features can serve as effective generative conditions, nor exploration of failure cases or transfer limitations

**Questions:**

please refer to the weaknesses

---

> ### Author Response · Authors · 2025-11-22
> **Official Rebuttal (1/2)**
>
> Dear Reviewer Z3r9,
>
> Thank you for your feedback. We would like to address your concerns one by one.
>
> **Q1: Limited Novelty, MetaQuery is similar to prior adapters**
>
> **A1:** As discussed in our related work, we agree that learnable queries are a standard design for adapters to extract and sample information (e.g., Flamingo [1], Q-Former [2]), and they are also used in unified models such as DreamLLM [3] and SEED-X [4]. However, MetaQuery is still novel in two key aspects:
>
> **How do queries aggregate information?**
>
> 1. Adapters like Q-Former in BLIP-2 [2], the Perceiver Resampler in Flamingo [1], or the projection layer in Next-GPT [6] are external modules that extract features from a frozen encoder via cross-attention. Their query tokens only see the encoder’s final outputs and cannot affect the internal computation of the backbone; they merely resample a static representation.
> 2. MetaQueries are closer to prompt tuning [8] on frozen LLMs. We append learnable queries directly to the input sequence and process them with the LLM’s native causal self-attention. Thus, MetaQueries become part of the context window throughout the entire forward pass. They aggregate not only static features, but also the dynamic reasoning trajectory of the LLM.
>
> **Who are generating images?**
>
> Even among methods that append learnable queries to the LLM input (e.g., DreamLLM [3], SEED-X [4]), our approach changes the direction of information flow from a *Push* paradigm to a *Pull* paradigm.
>
> 1. The *Push* Paradigm (Seed-X [4], Next-GPT [6], DreamLLM [3]): Prior works operate on a *Push* mechanism, **LLMs learns to generate images**. They train the LLM to generate image-aligned embeddings and *push* them to the generator. The LLM is forced to alter its output space to accommodate generation, which often requires expensive full-parameter training and risks catastrophic forgetting of text capabilities. An evidence of this forgetting risks is that all unified models in below Table achieves worse reasoning performance than T2I models (with frozen text encoders). This highlights our efforts to better transfer the capabilities of LLMs to image generation.
>
> 2. The *Pull* Paradigm (Ours - MetaQueries): In contrast, our method establishes a *Pull* mechanism, **diffusion models are generating images**. We treat the MLLM as a frozen, parametric knowledge base. Instead of forcing it to output visual signals, we use MetaQueries to actively probe and *pull* rich semantic features directly from its internal hidden states via self-attention. This effectively decouples generation from understanding. By *pulling* directly from internal representations rather than *pushing* an output, we preserve the MLLM’s original reasoning capabilities intact (Frozen Backbone) while retrieving denser, more nuanced visual conditions for the diffusion model.
>
> A natural concern is whether such frozen backbones, with only a few trainable queries, are sufficient for understanding complex prompts or encoding detailed information in images. In this work, our extensive experiments on text-to-image generation, image reconstruction, image editing, subject-driven generation, and knowledge-augmented generation demonstrate that the simple MetaQuery framework can already achieve strong performance with a lightweight training recipe. The overall goal is to deliver on the promise of unified models, but through a much simpler and more robust approach.
>
> **Q2: Theoretical Analysis of why or when frozen MLLM features work**
>
> **A2:** We appreciate the reviewer’s suggestion to deepen the analysis. While a formal mathematical proof is beyond the scope of this empirical study, we provide the following insights into why and when frozen MLLM features are effective. As shown in Table 2, freezing the MLLM achieves performance comparable to fine-tuning it in text-to-image generation tasks. However, Figure 2 reveals that frozen MLLM features result in some information loss during image reconstruction. In short, frozen MLLM features are sufficient for semantic-level alignment but cannot fully preserve every fine-grained details needed for image reconstruction. We have additional visual results of using frozen MLLM features for image reconstruction in Figure 8 and Figure 10, it works but we admit it is definitely not VAE-level.

---

> ### Author Response · Authors · 2025-11-22
> **Official Rebuttal (2/2)**
>
> **Q3: Exploration of failure cases or transfer limitations**
>
> **A3:** Thanks for asking. As shown in Table 4 and Table 6, we observe improvements over base diffusion models in text alignment, image fidelity, and, in particular, reasoning and world-knowledge-augmented generation when connected with powerful MLLMs. However, there is still considerable room for improvement.
>
> 1. For example, GPT-4o achieves a score of 0.80 on the WISE benchmark, significantly outperforming open-source models. Although base MLLMs can perform strong reasoning and answer complex questions, they often struggle to generate images with the same level of accuracy. This suggests that we still cannot fully transfer the reasoning capabilities of MLLMs to image generation.
> 2. There are also many failure cases in text alignment, as MetaQuery remains constrained by the limitations of current diffusion models. For instance, it cannot generate an image for prompts like "*a horse riding an astronaut*," which continue to pose challenges for text-to-image models.
>
> **Table1. Comparison on reasoning augmented generation on WISE**
>
> | **Model**             | **WISE** |
> | --------------------- | -------- |
> | **Dedicated T2I**     |          |
> | FLUX.1-dev            | 0.50     |
> | PixArt-Alpha          | 0.47     |
> | playground-v2.5       | 0.49     |
> | SD-v1-5               | 0.32     |
> | SD-3.5-large          | 0.46     |
> | **Unify MLLM**        |          |
> | GPT4o (evaled by [7]) | **0.80** |
> | Emu3                  | 0.39     |
> | JanusFlow-1.3B        | 0.18     |
> | Janus-Pro-1B          | 0.26     |
> | Janus-Pro-7B          | 0.35     |
> | show-o-512            | 0.35     |
> | vila-u-7b-256         | 0.31     |
> | MetaQuery-XL          | **0.55** |
>
> **Reference:**
>
> [1] Alayrac, J. B., Donahue, J., Luc, P., Miech, A., Barr, I., Hasson, Y., ... & Simonyan, K. (2022). Flamingo: a visual language model for few-shot learning. Advances in neural information processing systems, 35, 23716-23736.
>
> [2] Li, J., Li, D., Savarese, S., & Hoi, S. (2023, July). Blip-2: Bootstrapping language-image pre-training with frozen image encoders and large language models. In International conference on machine learning (pp. 19730-19742). PMLR.
>
> [3] Dong, R., Han, C., Peng, Y., Qi, Z., Ge, Z., Yang, J., ... & Yi, L. (2023). Dreamllm: Synergistic multimodal comprehension and creation. arXiv preprint arXiv:2309.11499.
>
> [4] Ge, Y., Zhao, S., Zhu, J., Ge, Y., Yi, K., Song, L., ... & Shan, Y. (2024). Seed-x: Multimodal models with unified multi-granularity comprehension and generation. arXiv preprint arXiv:2404.14396.
>
> [5] Tong, S., Fan, D., Li, J., Xiong, Y., Chen, X., Sinha, K., ... & Liu, Z. (2025). Metamorph: Multimodal understanding and generation via instruction tuning. In Proceedings of the IEEE/CVF International Conference on Computer Vision (pp. 17001-17012).
>
> [6] Wu, S., Fei, H., Qu, L., Ji, W., & Chua, T. S. (2024, July). Next-gpt: Any-to-any multimodal llm. In Forty-first International Conference on Machine Learning.
>
> [7] Yan, Z., Ye, J., Li, W., Huang, Z., Yuan, S., He, X., ... & Yuan, L. (2025). Gpt-imgeval: A comprehensive benchmark for diagnosing gpt4o in image generation. arXiv preprint arXiv:2504.02782.
>
> [8] Lester, B., Al-Rfou, R., & Constant, N. (2021). The power of scale for parameter-efficient prompt tuning. *arXiv preprint arXiv:2104.08691*.

---

### Author Response · Authors · 2025-12-04

Dear AC and Reviewers,

We sincerely appreciate your time and effort in reviewing our work. Although the reviewers were unable to participate in the discussion during the rebuttal phase, we believe our responses adequately addressed the concerns raised. Below, we provide a concise summary of the main concerns and our rebuttals:

**Similar to prior adapters such as Q-Former (Addressing Z3r9):**

MetaQuery operates directly with large frozen LLMs rather than small, trainable adapters. It is like prompt tuning the frozen LLMs, which is new for building unified multimodal models.

**Clarification on “Connecting two existing modules is not novel” (Addressing hJkF):**

We believe this is a misunderstanding. First, the MetaQuery design is orthogonal to the use of pretrained models, it is fully compatible with image generators trained from scratch, as in BLIP-3o. Second, MetaQuery is not about connecting LLMs with diffusion models. We aim to propose a different, significantly more efficient paradigm for building unified models:

* Rather than existing methods train LLMs to produce image-aligned embeddings, we keep the MLLM completely frozen and use learnable queries to extract information. This represents a fundamental shift in how models interact, our method does not ask LLMs to generate images.
* We clarify that a unified model should be defined by its functionality (understanding & generation) rather than a monolithic architecture. Our approach achieves this synergy by transferring LLMs' understanding capability to image generation without the common trade-off of degrading the MLLM's understanding capabilities.

**Connector is too large (Addressing NM2y, 7rvH, hJkF):**

A major concern was the size of the 24-layer transformer connector. We have implemented a lightweight MLP connector (only 10M parameters). With appropriate normalization, this MLP connector not only drastically reduces parameters (from ~2B to 10M), but also achieving even better performance.

Thank you again for your thoughtful review and feedback.

---

### Meta-Review · Area_Chair_Fb6i · 2026-01-06

**Summary:**

This paper received scores below the acceptance threshold, with shared major concerns regarding novelty, efficiency, theoretical analysis, and missing experiments. After reviewing both the paper and the rebuttal, the AC believes that the authors have adequately addressed the efficiency concerns. However, several important issues remain unresolved.

(1) Novelty & Theory. Unified multimodal models with learnable queries have been previously explored in works such as DreamLLM and SEED-X, as noted by reviewers Z3r9 and hJkF. The key concern that whether replacing a learnable MLLM with a frozen MLLM constitutes sufficient novelty to meet the bar of a top-tier conference remains. The authors attempted to justify this contribution through a newly proposed pull–push theoretical framework, which the AC believes is potentially interesting and warrants another round of review.

(2) Missing Experiments. Reviewer NM2y requested an ablation in which the MLLM’s output tokens are used directly as input; this request was not explicitly addressed in the rebuttal. Although Table 4 (“LLM last-layer embedding of input tokens”) may partially relate to this question, the current comparison may be unfair for several reasons: (a) MetaQuery introduces an additional 2B parameters while yielding only marginal improvements; (b) the baseline could likely be strengthened based on the authors’ new findings during rebuttal (e.g., the importance of LayerNorm discussed in response to NM2y Q1), which were not reflected in the baseline configuration. The AC therefore believes that a new, fairer experimental comparison is required. In addition, reviewer hJkF noted the absence of tasks that could more clearly demonstrate the advantages of MetaQuery, such as interleaved image–text generation.

Overall, due to these outstanding major concerns, the AC estimates that the final scores after rebuttal would be 4, 6, 6, and 4, which indicate a rejection. **The AC agrees with the reviewers and recommends rejection.** The submission would benefit from substantial revisions informed by the rebuttal and a further round of review at a future venue.

**Reviewer Concerns:**

| Reviewer | Addressed Concerns | outstanding Concerns |
|--------|------------------|------------------|
| Z3r9 | theory | Limited novelty. Similarity to DreamLLM and Seed-X. |
| NM2y | Connector size; efficiency | The concern in using MLLM's output tokens as direct input for the connector requires more clarification for authors. AC believes the required ablation might already be available in Table 4 if the reviewer asked for using LLM last-layer embedding as diffusion tokens (MetaMorph like). In another hand, if the reviewer asked for auto-regressively generating new tokens for the embeddings without the metaquery, the author could provide such ablation as it seems missing from the mauscript.|
| 7rvH | Attention mask; connector size;  |  AR vs diffusion. AC believes this is a big topic that is beyond the scope of this work though. |
| hJkF | Novelty | (1) Missing tasks to justify why MetaQuery, one possibility is the task of interleaved image-text generation as suggested, which is still missing from the current version. (2) Similarity to DreamLLM and Seed-X. |

**Reviewer Scores:**

| Reviewer | Initial Score | AC Estimated Score | AC Reason |
|--------|---------------|-------------------|-----------|
| Z3r9 | 4 | 4 | Novelty concerns remained |
| NM2y | 6 | 6 | Efficiency concern resolved |
| 7rvH | 4 | 6 | Major concerns in attention and connector sizes addressed |
| hJkF | 2 | 4 | Major concerns addressed. Reviewer mentioned: ``I will increase my score accordingly when I can''. Given the interleaved task is missing, the reviewer might increase to 4 most probably. |

The final estimated score 4, 6, 6, 4 indicates a rejection.

---

### Decision · Program_Chairs · 2026-01-26

Reject